# Small Extracellular Vesicles’ miRNAs: Biomarkers and Therapeutics for Neurodegenerative Diseases

**DOI:** 10.3390/pharmaceutics15041216

**Published:** 2023-04-11

**Authors:** Wei Qing Lim, Kie Hoon Michelle Luk, Kah Yee Lee, Nasuha Nurul, Sin Jade Loh, Zhen Xiong Yeow, Qi Xuan Wong, Qi Hao Daniel Looi, Pan Pan Chong, Chee Wun How, Sharina Hamzah, Jhi Biau Foo

**Affiliations:** 1School of Pharmacy, Faculty of Health and Medical Sciences, Taylor’s University, Subang Jaya 47500, Selangor, Malaysiasharina.hamzah@taylors.edu.my (S.H.); 2My CytoHealth Sdn. Bhd., Lab 6, DMC Level 2, Hive 5, Taman Teknologi MRANTI, Bukit Jalil, Kuala Lumpur 57000, Malaysia; 3National Orthopaedic Centre of Excellence for Research and Learning (NOCERAL), Department of Orthopaedic Surgery, Faculty of Medicine, Universiti Malaya, Kuala Lumpur 50603, Malaysia; panpanchong@um.edu.my; 4School of Pharmacy, Monash University Malaysia, Bandar Sunway, Subang Jaya 47500, Selangor, Malaysia; how.cheewun@monash.edu; 5Medical Advancement for Better Quality of Life Impact Lab, Taylor’s University, Subang Jaya 47500, Selangor, Malaysia

**Keywords:** extracellular vesicles, exosomes, miRNAs, neurodegeneration, stem cells

## Abstract

Neurodegenerative diseases are critical in the healthcare system as patients suffer from progressive diseases despite currently available drug management. Indeed, the growing ageing population will burden the country’s healthcare system and the caretakers. Thus, there is a need for new management that could stop or reverse the progression of neurodegenerative diseases. Stem cells possess a remarkable regenerative potential that has long been investigated to resolve these issues. Some breakthroughs have been achieved thus far to replace the damaged brain cells; however, the procedure’s invasiveness has prompted scientists to investigate using stem-cell small extracellular vesicles (sEVs) as a non-invasive cell-free therapy to address the limitations of cell therapy. With the advancement of technology to understand the molecular changes of neurodegenerative diseases, efforts have been made to enrich stem cells’ sEVs with miRNAs to increase the therapeutic efficacy of the sEVs. In this article, the pathophysiology of various neurodegenerative diseases is highlighted. The role of miRNAs from sEVs as biomarkers and treatments is also discussed. Lastly, the applications and delivery of stem cells and their miRNA-enriched sEVs for treating neurodegenerative diseases are emphasised and reviewed.

## 1. Introduction

Neurodegeneration refers to the death of neurons in the central nervous system (CNS) and peripheral nervous system (PNS). When CNS is affected, neurodegeneration causes Alzheimer’s disease (AD), Parkinson’s disease (PD), Huntington’s disease (HD), and others [1]. Around 50 million people worldwide have dementia, which is predicted to increase rapidly to 130 million people in the next 30 years. Most of these dementia cases are caused by AD. Neurodegenerative diseases will eventually lead to disability, institutionalisation, and mortality that requires high treatment costs of up to USD 1 trillion worldwide [2].

All neurodegenerative diseases are associated with functional neuron loss—progressive and irreversible. When neuronal structures deteriorate, there will be a gradual loss of cognitive skills (dementia) and motor skills (ataxia), leading to mental impairment and debilitation [3]. Some common mechanisms involved in all neurodegenerative diseases are oxidative stress, inflammatory responses, and the aggregation of abnormal misfolded proteins in selectively vulnerable neuron populations. Neurodegenerative diseases are challenging to treat because the damaged or dead neuron cells are naturally irreplaceable [3].

Neurodegenerative diseases cannot be completely cured, and the available treatments only help to alleviate pain, control symptoms, and enhance patients’ ability to move their limbs (Table 1) [4]. Various preclinical and clinical studies conducted 40 years ago revealed that cell therapy is the only rational and feasible strategy to regenerate neural tissues, mainly using stem cells. However, stem cell therapy has disadvantages as its therapeutic molecules cannot pass through the blood-brain barrier (BBB) effectively and are detrimental to patient safety [4]. Mesenchymal stem cells (MSCs) are adult multipotent progenitor cells that can renew themselves and undergo differentiation into various mesodermal cell lineages, including bone, cartilage, and adipose tissues. However, MSCs have several other disadvantages in clinical settings, which include a greater risk of transmitting infections and their potential to differentiate decreasing in long-term culture [5].

Furthermore, after injecting MSCs into the body intravenously, MSCs undergo a pulmonary first-pass effect leading to the accumulation of 50% to 80% of MSCs in the lungs due to MSC’s huge molecular size of 20 μm to 30 μm diameter, which is larger than circulating immune cells and lung capillaries [11]. Besides this “lung-trapping” effect, for all delivery routes such as intravenous, intratracheal, intraperitoneal, and intranasal, MSCs could only survive less than 24 h in the respiratory tract, making it difficult for MSCs to be delivered to other organs. For instance, the effectiveness of using MSCs to treat neurodegenerative diseases is much restricted as less than 1% of MSCs are detected in the brain following administration [12].

Due to the limitations of MSC treatment, various clinical studies are being conducted based on small extracellular vesicles (sEVs) extracted from MSCs and enriched with miRNAs to treat neurodegenerative diseases more safely and effectively [13]. Studies have found that after systemic transplantation, MSCs are quickly excreted from the body but their pharmaceutical effects are usually maintained. This indicates that the molecules responsible for the therapeutic effect are secretomes of MSCs, especially sEVs. Indeed, in order to deliver therapeutic molecules to the brain more efficiently by crossing the BBB, sEVs are used as an alternative strategy to treat neurodegenerative diseases owing to their tiny molecular size of 30 nm to 100 nm [14].

Due to their ability to mediate intercellular communication, MSCs’ sEVs may affect the target cells by interacting with their receptors or by delivering biologically active cargo molecules such as proteins, mRNA, and microRNAs (miRNAs). Intercellular communication occurs when the target cells receive or engulf sEVs via endocytosis or phagocytosis [15]. Non-coding RNAs with 17–21 nucleotides called miRNAs are among the most significant sEVs’ cargoes. Canonically, a base pairing of nucleotides 2–7 of miRNAs (the seed) to mRNA 3′ untranslated regions (UTRs) prevents the translation of mRNAs [15]. Various studies have reported that MSCs’ sEVs show a promising therapeutic potential for neurodegenerative diseases [16]. MSC-sEVs can prevent the apoptosis and inflammation caused by microglial cells and demyelination and suppress oxidative stress [17]. Hence, the functional deterioration of neurons can be prevented. Additionally, MSC-sEVs play a vital role in the regeneration of neurons via four mechanisms of action: neuroprotection, neurogenesis, neuromodulation, and angiogenesis [18]. Evidence shows that MSC-sEVs can preserve myelin sheaths and synapses [19]. In this article, the pathophysiology of various neurodegenerative diseases is reviewed. The role of miRNAs from sEVs as biomarkers and treatments is also discussed. Lastly, the applications and delivery of stem cells and their miRNA-enriched sEVs for treating neurodegenerative diseases are emphasised and reviewed.

## 2. Pathophysiology of Neurodegenerative Diseases

### 2.1. Alzheimer’s Disease

Alzheimer’s disease (AD) was first discovered in 1906 by the German psychiatrist Alois Alzheimer and is now hailed as one of the leading causes of dementia globally [20]. Patients with dementia experience progressive deterioration in performing mental and motor tasks. They often begin to display worrying emotional responses, which can take a toll on their caregivers in the long run. Post-mortems of patients’ brains reveal an accumulation of Aβ plaques in the extracellular region and neurofibrillary tangles (NFTs) that originate due to hyperphosphorylated microtubule-associated τ intracellularly. β-secretases (BACE1) and γ-secretases cleave the amyloid precursor protein (APP) to produce insoluble Aβ fibrils [21], which then oligomerise and diffuse into synaptic clefts to disrupt synaptic signalling. Consequently, polymerisation into insoluble amyloid fibrils leads to aggregation into plaques. The polymerisation process activates protein kinases such as GSK3β and CDK5, which promote the hyperphosphorylation of the τ protein [22]. The τ proteins then oligomerise, contributing to the dissociation of microtubule subunits. The microtubules fall apart and accumulate into τ filaments, which further aggregate into insoluble NFTs [23]. This negatively affects cell-to-cell communication and signal processing, leading to neuron cell death and increased reactive oxidative stress [24].

Furthermore, this triggers the infiltration of microglia, which act as phagocytes and are vital in neuronal plasticity and synapse remodelling [25]. Aβ plaques activate receptors such as CD36, SCARA1, α_6_β_1_ integrin, CD14, toll-like receptors, and CD47 on the microglia, leading to the activation and induction of microglia neuroinflammation through the release of pro-inflammatory cytokines and chemokines [26]. Microglia induce the endocytosis of Aβ oligomers and NFT fibrils through neprilysin and insulin-degrading enzymes through this cascade. However, AD can be exacerbated due to an excessive or insufficient inflammatory response [27]. The triggering receptor expressed on myeloid cells 2 mutation or glial cell activation prolongs the neuroinflammation process and leads to reactive oxygen species’ overexpression that further damages neuronal cells. Eventually, the clearance of waste proteins is impaired [28].

### 2.2. Parkinson’s Disease

Parkinson’s disease (PD) is the second most common neurodegenerative disease globally. It is characterised by tremor, rigidity, and bradykinesia with late-stage postural instability. Although PD is not fatal, patients suffer a drastic decrease in quality of life as retardation in mobility progresses [29]. They tend to experience falls more frequently and develop aspiration pneumonia contributing to life-threatening infections. Despite being a prevalent disease, the cause of the disease is largely unknown, with some speculating that smoking, pesticide exposure, and familial history are associated risk factors [30].

Chiefly, the pathological presentation of PD begins in the substantia nigra, where declining dopaminergic neurons are observed. This is accompanied by a severe deficiency of dopamine concentration in the striatum. α-synuclein is found naturally in the brain and presents as a tertiary protein in unfolded form. It can be folded into α-helical structures at the N-terminus upon interaction with phospholipids [31]. Pathological development arises when α-synuclein is folded into a β-sheet-rich amyloid-like structure, possibly due to serine 129 phosphorylation, ubiquitination, and C-terminal truncation resulting in misfolding and aggregation [32]. Such misfolded proteins can present as unfolded monomers, soluble oligomers, protofibrils, and insoluble fibrils [33]. Among the four types, the oligomers present the largest threat due to their capability of “seeding” and accelerating α-synuclein accumulation. Researchers also note that α-synuclein interacts with the TOM20 receptor on the mitochondrial membrane, leading to the damage of complex-I, a vital component of the electron transport chain [34]. This results in mitochondrial dysfunction through impaired mitochondrial protein import machinery, reduced respiration, and excessive production of reactive oxygen species [35]. This theory is further supported by a deficiency of mitochondrial complex-I in patients’ brains, skeletal muscles, and platelets. Furthermore, the exposure of individuals to neurotoxins such as MPTP, rotenone, and paraquat resulted in the inhibition of the same complex. Lastly, the PINK1 and parkin genes commonly found in hereditary PD patients have also been found to regulate the removal of dysfunctional mitochondria (Figure 1) [36].

Apart from that, dysfunctional protein clearance systems (the ubiquitin-proteasome system and autophagy-lysosome system) also play a significant role in the pathophysiology of PD. Typically, the ubiquitin-proteasome system (UPS) breaks down soluble misfolded α-synuclein through ubiquitin tagging and transportation to the proteasome for degradation. In contrast, the activity of UPS and proteasome component levels (PA700 and PA28 found in the 20S proteasome α-subunit) are reduced in the brain and peripheral blood mononuclear cells in PD patients [37]. The parkin and UCH-L1 genes in diseased patients have also been determined to regulate the ubiquitin-encoding E3 ubiquitin ligase and ubiquitin C-terminal hydrolase [38]. The autophagy-lysosome system removes abnormal proteins through macroautophagy, microautophagy, and chaperone-mediated autophagy. In PD patients, although the autophagosome marker LC3-II was found to be increased, the vital proteins of lysosomal membranes (LAMP1 and LAMP2A) and several molecular chaperones from the heat-shock protein family (hsc70 and hsp35) were decreased [39]. Again, the parkin and PINK1 genes were also found to regulate the autophagic turnover of mitochondria, confirming that altered protein clearance systems play a role in the development of the disease. Gradually, the accumulation of α-synuclein directly triggers microglial activation (similar to AD) and initiates inflammatory processes [40]. This is confirmed through post-mortem studies demonstrating evidence of microglial and complement activation, T-lymphocyte infiltration, and increased concentrations of pro-inflammatory cytokines [41].

### 2.3. Huntington’s Disease

Huntington’s disease (HD) is a rare autosomal dominant condition with a prevalence of 0.4/100,000 in Asia and 5.70/100,000 in the Western population. It is characterised by progressive movement disorder and cognitive decline. Patients experience motor defects such as chorea (resulting in involuntary muscle movement) and loss of coordination. Obsessive-compulsive disorder, depression, and psychosis are also common among sufferers. The disease is currently incurable, and patients often succumb to the disease roughly 20 years after disease onset [42].

The consensus among researchers infers that the abnormal expansion of CAG repeats in the HTT gene encoding huntingtin causes HD. The protein contains a polyglutamine tract encoded by uninterrupted CAG trinucleotide repeats in the first exon of HTT. Healthy individuals possess the wild-type allele that contains up to 35 CAG repeats, while the pathogenic allele carries expansions of 36 or more repetitions [43]. Huntingtin has been revealed to be crucial in neurogenesis, synapse connectivity, and neuron cell survival by acting as a protein scaffold and transcriptional regulator. Therefore, a diseased state results in general brain shrinkage, degeneration of the striatum, and the loss of efferent medium spiny neurons [44].

As with other neurodegenerative diseases, HD is characterised by the presence of aggregates in the brain. The long CAG repeats may be cleaved by caspases and calpains to form toxic N-terminal fragments, which subsequently form β-sheets when held together by hydrogen bonds, ultimately assuming an amyloid structure. Gradually, the aggregates may sequester ubiquitin, proteasome subunits, chaperones, transcriptional factors, or wild-type alleles which induce neurotoxic effects, leading to an impairment in the ubiquitin-proteasome system. Indeed, the aggregates show abnormal axonal transport of the autophagosome-lysosome system, contributing to inefficient autophagosome-lysosome fusion and decreased degradation of the aggregates [45].

Moreover, scientists have discovered transcriptional disruption in HD patients. Mutant huntingtin has been shown to interact with regulators of transcription such as p53, cAMP response element-binding (CREB) protein, and CREB-binding protein (CBP), which are essential for cell growth and survival [46]. Furthermore, mutant huntingtin also interacts with several other molecules and affects their underlying molecular mechanisms, for instance: PGC-1α (peroxisome proliferator-activating receptor-γ coactivator-1 α), which is required for energy metabolism; Sp1 and its coactivator TAFII130, which affect the transcription of genes such as the D2 dopamine receptor; cystathionine γ-lyase, the biosynthetic enzyme for cysteine; and lastly, BDNF, a pro-survival factor that enhances the survival of striatal neurons [44]. Many also believe that huntingtin disrupts the epigenetic landscape through deregulation of histone modification by binding to histone acetyltransferase domain in CBPs to disrupt histone acetylation, induce chromatin structural modifications in downregulated genes, and change the DNA methylation of several altered genes [47]. Finally, miRNA deregulation has been shown to exacerbate HD, as cleaved CAG RNAs silence specific genes. Notably, the cleaved RNAs are potentially neurotoxic through Ago-2-mediated gene silencing of CTG-containing genes [48].

A combination of aggregates and gene silencing ultimately leads to neuronal and synaptic abnormalities. The aggregates delay axonal transport due to Huntington-associated-protein-1 disruptions, resulting in a failed delivery of GABA(A), which inhibits synaptic excitability and leads to neuronal cell death (Figure 2). The disruption in the trafficking of NMDA receptors in striatal neurons may cause an imbalance between synaptic and extrasynaptic NMDA receptors. Consequently, excitotoxicity happens either due to increased glutamate or impaired uptake and clearance [49]. Ultimately, this leads to defects in ATP production, Ca^2+^ buffering capacity, and mitochondrial apoptosis. Overexpression of the myeloid lineage-determining factors PU.1 and CCAAT/enhancer-binding proteins results in a pro-inflammatory cascade contributing to neuroinflammation. The inhibition of NF-κB signalling in the peripheral immune system also increases peripheral inflammatory responses [50].

### 2.4. Amyotrophic Lateral Sclerosis

Amyotrophic lateral sclerosis (ALS) is an agonising and rapidly progressive neurodegenerative disease characterised by motor neuron degeneration and progressive muscle weakness. Most patients receive their initial diagnosis between the ages of 40–70 and will succumb to respiratory failure within 3–5 years. Currently, researchers are unsure of the exact mechanism contributing to ALS; hence, therapeutic options remain limited. Three theories have been proposed concerning ALS pathogenesis: the dying forward hypothesis, the dying back hypothesis, and the independent hypothesis (Figure 3) [51]. In the dying forward hypothesis, it is theorised that ALS results in anterograde corticomotor neuron degeneration. Dysfunctional astrocytic excitatory amino acid transporter 2 is implicated in reduced glutamate uptake at the synaptic cleft. Ultimately, degradation of the anterior horn cells ensues due to excitotoxicity. In the dying back hypothesis, ALS begins within the neurofilaments. Mutations in the TDP-43, c9orf72, and fused in sarcoma (FUS) genes result in disruptive RNA metabolism, abnormalities of gene translation, and eventually, the formation of intracellular neuronal aggregates [52]. Aside from that, disruption of the neurofilament-L (NF-L) gene in SOD1 also increases oxidative stress and mitochondrial dysfunction, forming aberrant perikaryal and axonal aggregates of neurofilaments [53]. Neurofilament-dependent slowing of slow axonal transport mediates damage to the cargoes and deprives axons of essential nutrients. Ultimately, this leads to motor axon degeneration. The independent degeneration hypothesis argues that upper and lower motor neuron degeneration occurs independently. Separately, it is found that microglia-induced neuroinflammation is noted in ALS patients [54].

## 3. Applications of Small Extracellular Vesicles’ miRNAs

Numerous published papers have highlighted the cargo of sEVs, including proteins, RNA, DNA, lipids, etc. [55]. Among all these cargoes, sEVs’ miRNAs have gained the spotlight in the last decade for both disease identification and treatment. It has been shown that sEVs synthesized and released by brain nerve cells can cross the BBB and present in blood and peripheral fluids [56]. Hence, temporal changes in the expression of sEVs’ miRNAs during the progression of neurodegenerative diseases can be identified, which enhances their ability to act as diagnostic tools [57]. In terms of treatment, MSCs’ sEVs-miRNAs have been widely reported to improve the pathophysiology of neurodegenerative diseases in preclinical studies. Although challenges are faced in the isolation of sEVs, consistent characterisation methods and techniques to increase genomic and proteomic yield are being rapidly improved to improve their usability in clinical practice [58].

### 3.1. Disease Biomarkers

sEVs’ miRNAs are gaining traction from multiple researchers as a reliable biomarker for neurodegenerative diseases. Over 2000 types of miRNAs have been identified that influence the gene regulation of essential biological pathways such as neurogenesis and neuronal growth, cell communication, and cell death [55]. The brain is highly enriched in miR-7, miR-9, miR-23, miR-125 a-b, miR-128, miR-137, miR-132, and miR-139. There are also various brain specific miRNAs such as miR-101, miR-191, let-7 g, miR-134, miR-181a-b, miR-135, miR-107, miR-let-7c, let-7a, miR-29a, and miR-124 [59]. Due to their capacity for intercellular communication, distant cells can pick up sEVs’ miRNAs that reflect the current functional status of the cell and alterations that contribute to disease pathogenesis and progression. Such alterations occur long before symptoms bud and an official diagnosis is determined, making them a valuable diagnostic tool for diseases [60]. Moreover, they can differentiate between diseases with similar symptoms, such as AD and PD, further branching into the identification of subtypes of the diseases [61]. To illustrate, a comparison of miRNA-135a, miRNA- 193b, and miRNA-384 levels in serum-derived sEVs from individuals with AD, PD, and vascular dementia can be made by observing the area under the curve from ROC curve analysis, qRT-PCR analysis, or miRNA PCR array analysis. Lastly, novel technology such as next-generation deep sequencing enables the identification of sEVs’ miRNAs free of contaminant RNA from blood samples, accelerating the possibility of using sEVs’ miRNAs as biomarkers. SEVs isolated from the biological fluid are protected from degradation, allowing for the easy identification of miRNAs [62].

### 3.2. Delivery System

As sEVs can carry diverse cargo, multiple miRNAs can be shuffled into a single preparation to treat patients with chronic medical conditions requiring multiple medications. Hence, patient compliance might be improved. Moreover, the risks associated with polypharmacy can be reduced drastically. Nowadays, scientists have successfully formulated intranasal dosage forms that allow penetration of sEVs’ miRNAs across the BBB, indicating that sEVs’ miRNAs could be available over the counter for patients [63]. To know more about the current status of the sEVs delivery system that can be used for neurodegenerative diseases, it is recommended to read our previous publication in which the intravenous delivery system has been critically reviewed [64]. The possible routes of administration are also critically reviewed in the later section.

### 3.3. Disease Treatment

miRNAs possess both pathogenetic and protective effects towards neurodegenerative diseases. Hence, sEVs’ miRNAs present a possibility for use as treatments for disease. Selective modification of the sEVs’ cargo or the biological production of protective sEVs’ miRNAs can activate target site receptors directly. The crossover of sEVs’ miRNAs through endocytosis enables them to deliver therapeutic miRNAs to intracellular components such as the cytoplasm and nucleus. Finally, modification of the sEVs with cell surface proteins allows for specific brain targeting (Figure 4) [65].

MSC-sEVs have been found to promote cell proliferation by transferring miR-17-92 clusters to the distal axons of cortical neurons. sEVs enriched with miR-124 from human CSF were able to promote neurogenesis in neural stem cells [66]. Yang et al., injected 12 μg (about 500 miRNAs per sEV) into stroke-induced mice weighing 22–25 g, in which the team observed neurogenesis [67]. Lastly, miR-132 can maintain the integrity of the BBB, postulating its significance in neurovascular communication and homeostatic functions. This is demonstrated by injecting 0.8 nmol miR-132 in 4 μL phosphate buffer solution into C57BL/6 mice weighing 20–25 g [68].

In patients experiencing a stroke, the transfer of miR-133b and miR-17-92 clusters through MSC-sEVs promoted plasticity and functional recovery. miR-133b inhibits the connective tissue growth factor and helps to encourage axonal growth by reducing microglial scarring. It also acts on the RhoA protein to promote spinal cord recovery. This effect was demonstrated by Yu et al., (2011), who injected 100 ng/μL of green fluorescent and red fluorescent protein–mRNAs solution containing 10 μM of miR-133b into adult zebrafish (approximately 2.5 cm) [69]. Furthermore, miR-17-92 inhibits PTEN gene expression in neurons, activating the PI3K/Akt/mTOR pathway. This was demonstrated by Xin et al., by injecting 100 μg/0.5 mL of phosphate buffer solution containing sEVs–miR17-92 into mice weighing 270–300 g [70]. Simultaneously, NMDA receptor activity was shown to be downregulated, while GABAA receptor activity is upregulated. This ultimately promoted axonal growth and accelerated nerve repair accompanied by reduced excitotoxicity [71].

Aside from that, Ma et al., showed that miR-181a targets Kruppel-like factor 6 and increases the permeability of the blood–tumour barrier, which allows enhanced delivery of therapeutics to treat gliomas [72]. Unsurprisingly, a downregulated level of miR-181a was observed in 80 tissue samples from glioblastoma patients [72]. Yu et al., found that sEVs–miRNA-199a inhibits glioma progression by downregulating ArfGAP with the GTPase domain, ankyrin repeat, and PH domain 2 [73]. The team also demonstrated that human MSCs transfected with miR-199a are able to inhibit tumour growth through the delivery of sEVs [73]. Interestingly, in vivo studies showed that miRNA-199a increased the chemosensitivity of glioma cells to temozolomide by inhibiting the K-RAS signalling pathway [74].

Lastly, Wang et al., transfected 10 μL of miR-146a onto dorsal root ganglia neurons of diabetic mice to prove that downregulation of miR-146a induced cell apoptosis through activation of IRAK1, TRAF6, and caspase-3 [75]. Elevations in IRAK1 and TRAF6 are known to impair wound healing and induce peripheral neuropathy in patients with type two diabetes. Therefore, the study concluded that maintenance of miR-146a levels achieves possible therapeutic effects in diabetic patients [76].

## 4. Small Extracellular Vesicles’ miRNAs for Neurodegenerative Diseases

### 4.1. Small Extracellular Vesicles’ miRNAs

sEVs’ miRNAs, particularly those involved in neurodegenerative illnesses, are definitively being shown to play a critical role in disease progression. Three main mechanisms explain how miRNAs contribute to neurodegenerative diseases: targeting regulatory-related gene mRNA for protein translation inhibition or protein degradation, taking part in neuroinflammation by directly interacting with a toll-like receptor or controlling its mRNA expression, and producing distortion in miRNAs’ formation [77]. In various neurodegenerative diseases, the expression of miRNA changes, and some of these changes have been linked to disease progression.

In addition, when the donor cells release the sEVs, miRNAs will be transmitted to the targeted cells via three distinct methods: fusion with the plasma membrane, endocytosis into the cytoplasm, and exhibiting receptor–ligand interactions for binding to target cells (Figure 5) [78]. Additionally, it has been discovered that neurons can indirectly control astrocyte protein expression by delivering miRNAs to them via sEVs. The above elaboration proposed that neurons might regulate the protein expression of astrocytes while the secretion of sEVs’ miRNAs takes place [77]. On the other hand, miRNAs are also involved in the mRNA loading pathway into sEVs. Through a mechanism controlled by miR-1289, certain regions in the 3′ UTR of mRNA with around 25 nucleotides and a CTGCC motif within a stem-loop structure operate as a sorting sequence to sEVs [79,80]. With the aid of Rab5 immuno-localisation, Wnt3 protein enters the cell and carries out its role to modulate the three main Wnt signalling pathways: the β-catenin/GSK3 (glucogen synthase kinase 3) pathway, the planar cell polarity pathway, and the Wnt/Ca^2+^ pathway in the recipient cells [80].

### 4.2. Small Extracellular Vesicles’ miRNAs and Alzheimer’s Disease

Numerous studies have shown that sEVs reduce the signs and symptoms of AD. However, more research into precise molecular pathways is still needed [81]. sEVs made by hypoxia-pretreated MSCs showed enhanced miR-21 expression, suggesting that miR-21 can improve mice’s cognitive impairments and prevent the clinical signs of AD [82]. sEVs’ capacity to remove beta-amyloid was also revealed by Zhi-You Cai et al., (2018) [83]. Since sEVs–neprilysin may break down beta-amyloid, MSC-sEVs encapsulating small molecule therapies can effectively target beta-amyloid [84]. The inhibition of miR-125b-5p was found to have neuroprotective benefits against oxidative stress, as it decreased reactive oxygen species levels and mitochondrial membrane potential [85]. The result suggested that miR-125b-5p may be a new regulator of AD development and serve as a therapeutic target for AD treatment.

The hypothesis that these vesicles could be a promising source of APP cleavage was raised by the discovery of secretases (γ-, α-, and β-secretases), enzymes implicated in the proteolytic cleavage of APP, in isolated sEVs from an AD animal model. Indeed, it was discovered that APP undergoes cleavage within sEVs and is found in sEVs from human and mouse AD model brains [86]. Moreover, findings from Yuyama et al., (2014) supported the sEVs’ mentioned “clearance role” through the decrease in Aβ and Aβ-mediated synaptotoxicity [87]. The neuroprotection effects of MSC-sEVs against Aβ-induced neuronal damage include decreased extracellular Aβ oligomer level, secretion of sEVs containing antioxidant enzymes such as catalase, and paracrine action via the extracellular release of anti-inflammatory cytokines and trophic factors including IL-6, IL-10, and VEGF [88]. Additionally, after being incubated with sEVs, β-amyloid oligomers (Aβo) were restricted to their surface structure by interacting with proteins, such as the prion protein (PrPC). In fact, PrPC performs a dual function: a protective one, encouraging the production of amyloidogenic fibrils and preventing the neurotoxic complications caused by Aβo or its neurotoxic action on the neuronal surface of the Aβ receptor by capturing Aβ in sEVs by superficial recognition [89]. sEVs-targeting therapeutic strategies may therefore offer promising clinical results for AD.

### 4.3. Small Extracellular Vesicles’ miRNAs and Parkinson’s Disease

Xin et al., (2012) reported that the growth of neurons was stimulated upon the transfer of miR-133b by MSC-sEVs [90]. Xin et al., (2017) discovered miR-17-92 clusters with neurogenesis activity in MSC-sEVs, stimulating oligodendrogenesis and enhancing neuronal function [70]. miR-34a was revealed to play a role in the neurotoxic pathways of PD-associated neurotoxins, namely paraquat, rotenone, and 6-hydroxydopamine (6-OHDA) [77]. The increase in miR-34a expression induced by 6-OHDA and inhibition of NRF2 in cells was reversed by the antioxidant Schisandrin B. In a 6-OHDA mouse PD model, Ba et al., (2015) discovered that lentiviral-mediated miR-34a expression could reverse the behavioural improvement affected by Schisandrin B [91]. Injected sEVs have been demonstrated to prevent gait impairments caused by 6-OHDA, in addition to improving motor function due to the normalised activity and expression of striatal tyrosine hydroxylase (TH) [92]. The release of miR-34a from astrocytes and its delivery via sEVs can augment the sensitivity of dopaminergic neurons to neurotoxins by targeting Bcl-2 in a PD model [93]. Chen et al., (2019) reported in a study that oxidative-stress-induced apoptosis can be reduced by upregulation of miR-34a [94]. Jiang et al., (2019) revealed that an increase in oxidation resistance 1 (OXR1) was observed upon upregulation of sEVs’ miR-137, thus providing a neuroprotective effect against oxidative stress in PD mice [95].

Meanwhile, the Let-7 family is responsible for neurodegeneration via the activation of toll-like receptor 7 (TLR7), in addition to being discovered to downregulate the effects of leucine-rich repeat kinase (LRRK2) functional mutations, which play a role in PD pathogenesis [77]. The cleavage of extracellular α-synuclein, a crucial factor in the development of PD, by the MSC conditional media was shown in several studies to have neuroprotective effects. For instance, extracellular α-synuclein breakdown using in vitro and in vivo model tests demonstrated that matrix metalloproteinase 2 (MMP-2) in the MSC-secretome decreased insoluble-synuclein oligomers and resulted in an improvement in neuronal survival in PD diseases [96].

### 4.4. Small Extracellular Vesicles’ miRNAs and Huntington’s Disease

sEVs affect the nervous system to regulate mutant Huntington gene (mHtt) aggregation, mitochondrial dysfunction, cell death, and cell viability in HD [97]. sEVs secreted by MSCs are vital for relieving HD phenotypes, which upregulate phosphorylated CREB and PGC-1 and expedite non-apoptotic protein levels [98], in particular alleviating mHtt aggregation in R6/2 mouse neurons. Thus, sEVs-carried mHTT propagation is thought to be a novel mechanism for HD pathology, presenting a potential therapeutic target for reducing this neurodegenerative disease [99]. An increase in cyclin A2 expression is observed with the decrease in miR-124 expression, suggesting that the expression of cyclin A2 is controlled by miR-124, indicating its role in cell cycle dysregulation in HD cell models [100]. Recent research has sought to use sEVs’ miR-124 to treat HD symptoms in animal models. The viability of sEVs-based miR-124 in an HD model was confirmed, even though the results did not demonstrate an improvement in the symptoms of HD animal models [101]. Given that oxidative stress might change the expression levels of miRNAs, there are not enough studies that focus explicitly on the connection between miRNAs and oxidative stress in HD. More research is required to determine whether the interaction between miR-124 and oxidative stress is significant in the pathophysiological process of HD [77,102]. Additionally, it has been reported that miR-124 expression is decreased in HD patients and can increase the expression of the neuron-restrictive silencing factor, which suppresses the expression of brain-derived neurotrophic factors. This finding raises the possibility that abnormal miR-124 expression is a major factor in the pathogenesis of HD [77].

### 4.5. Small Extracellular Vesicles’ miRNAs and Amyotrophic Lateral Sclerosis

Rizzuti et al., (2018) established the importance of miR-34a in neurodegeneration and ALS. Sirtuin 1 (SIRT1), one of miR-34a’s targets, is a factor that protects against oxidative stress-induced apoptosis [103]. In the ALS in vitro model, elevated SIRT1 is also seen, in addition to miR-34a downregulation [103]. This finding implies that inhibiting miR-34a may protect against oxidative stress-induced apoptosis in ALS via boosting SIRT1 expression. Furthermore, epigenetic modifications regulate miR-34a expression through the demethylation of the promoter region of the miR-34a gene [77]. According to Wang et al., (2017), miR-142-5p suppression can activate Nrf2, which then blocks oxidative stress and cell damage through the OGD/R pathway. Validating the involvement of miR-142 in ALS would help us to better understand the pathogenesis and progression of ALS, given the significance of inflammation and oxidative stress in the disease [77].

Recent studies suggest that remyelinating potential is demonstrated in sEVs [104]. The downregulation of IFNγ would stimulate the secretion of sEVs containing remyelinating miRNA species by dendritic cells, forming a new compact myelinating area in the lysolecithin-induced demyelinating hippocampal slice [104]. sEVs may contain miRNA clusters effective in myelin production (miR219 and miR-17-92) and anti-inflammatory effects (miR-27a-b and miR-145). miR-219 is responsible for inhibiting platelet-derived growth factor receptor alpha, promoting cell proliferation while prohibiting differentiation. miR-219 also inhibits ELOVL7, which is responsible for modulating lipid metabolism to increase myelin basic protein and actively participating in myelination, hence solidifying its significant role in the remyelination of samples of demyelinated hippocampus tissue by sEVs [105]. In a related study, administering environmental enrichment to older animals increased the miRNA-219 levels within serum-derived sEVs, which was sufficient for producing myelinating oligodendrocytes [104], reinforcing its significant role in the remyelination of samples of demyelinated hippocampus tissue by sEVs [105]. The miRNAs involved in various neurodegenerative diseases are summarised in Table 2.

### 4.6. Roles of miRNAs and Proteins in sEVs

Among the various EV kinds described thus far, including microvesicles, microparticles, ectosomes, shedding particles, and apoptotic bodies, the sEVs are the most clearly defined. The sEVs’ membrane is similar to a cell in that it is rich in signalling molecules and surface antigens. Besides that, it is also said to include both proteins and genetic material, but no organelles such as the nucleus or mitochondria exist [115]. miRNAs play essential roles in gene expression, especially in post-transcription regulation. Since the cancerous cells can release miRNA species that can be used as diagnostic indicators, extracellular miRNAs have been studied to become potential circulating biomarkers for certain cancers and diseases [116]. The recent sEVs preparations that contain miRNAs raise the possibility that these vesicles function as a form of intercellular communication and as a means of transporting miRNAs across cells. One of the intriguing theories states that >60% of all mRNAs are regulated by miRNAs. Therefore, the activity of target mRNAs can be bound and inhibited by miRNAs [117]. Epstein–Barr virus-infected cells were the first to exhibit the intercellular transfer of miRNAs via sEVs that led to functional activity in recipient cells, where viral miRNAs were transferred to uninfected recipient cells, as well as to COS-7 cells and metastatic prostate cancer cells. Numerous MSC-sEVs-mediated cellular processes, including angiogenesis and anti-angiogenesis, immunomodulation, anti-apoptosis, and anti-fibrosis, have been linked to MSC-sEVs’ miRNAs. For example, sEVs’ miR-146a was up-regulated and enhanced macrophage polarisation to M2 macrophages in MSCs pre-treated with interleukin-1, which in turn attenuated inflammation and improved survival in septic mice [118].

Several groups have covered the proteome of MSC-sEVs to date. Meanwhile, there are about a thousand proteins known thus far. The proteome’s relationship to biological processes was mapped, and it became clear when those proteomes are known to play a significant role in crucial biological processes, including cellular communication, cellular structure, inflammation, sEVs biogenesis, development, tissue repair and regeneration, and metabolism [119]. Proteins in MSC-sEVs can affect a few biological processes involved in disease pathogenesis or tissue repair and regeneration, similar to miRNAs in MSC-sEVs. The comparison of the proteins’ and MSC-sEVs–miRNAs’ ability to elicit physiologically relevant activity is clear, as miRNAs are unlikely to be present in the proper structure or concentration. However, a normal sEVs dose does not include enough pre-miRNAs for a biologically meaningful reaction. According to the study by Chevillet et al., (2014), only a tiny fraction (3%) of cancer-related biomarker miRNAs were found in the recovered classical sEVs isolated using differential ultracentrifugation-based techniques. Therefore, this study suggests that these well-established biomarkers are present in plasma and serum in various physical forms, even though it does not directly address the diagnostic value of these sEVs [117]. Meanwhile, proteins in a typical therapeutic MSC-sEVs dose, as demonstrated by the example of glycolytic enzymes to create ATP, show their capacity to elicit a physiologically relevant response. Therefore, a protein-mediated mechanism of action stands out as the more likely route of the sEVs’ effect. However, it should be highlighted that structural proteins might not elicit a biological reaction and the ability to evoke a physiologically meaningful response is mainly attributable to the catalytic activity of enzymes [119].

## 5. Administration Route of Small Extracellular Vesicles

The efficacy and efficiency of delivering neuro-specific sEVs remain significant problems. This is because the administered therapeutic sEVs may be identified by immune cells, leading to subsequent destruction by the reticuloendothelial system (RES). It is also a consensus that the BBB is a major barrier to developing CNS-targeting therapeutics [120]. The route of administration influences its distribution and its therapeutic and biological effects. The common administration routes in neurodegenerative disease treatments are intravenous (IV), stereotactic, oral, and nasal.

### 5.1. Intravenous Injection

Currently, sEVs are mainly administrated intravenously [120]. A recent study found that intravenously injected sEVs could travel to the brain and enhance cognitive capabilities due to reduced plasma Aβ levels and normalised cytokine levels [121]. Furthermore, in another study, parkinsonian mice were injected with catalase-containing sEVs; this intervention demonstrated a protective effect towards SNpc neurons in mice [122]. Although the IV route shows promising effects, its scalability is limited by a variable half-life ranging from minutes to hours due to macrophage clearance during circulation in blood [123].

### 5.2. Stereotactic Injection

The stereotactic injection is performed by dispersing or dissolving the therapeutic component in a solvent and injecting the mixture intrathecally [124]. The stereotactic injection is more advantageous compared to the IV route due to its site-specificity and superior stability. Most importantly, the side effects are reduced due to a lower dose retention time [125]. To illustrate this, PPSweInd transgenic (APP) mice were injected with biotinylated sEVs in the right hippocampus. The results show potential for AD therapy, as the sEVs successfully entrapped Aβ proteins, improving the proteins’ clearance through immunological processes [87]. As with any pharmacological therapy, further improvements are required to improve therapeutic outcomes. The challenges associated with stereotactic injection are mandatory imaging guidance to locate the target position, high expertise requirement, and the potential for severe damage to the brain in cases of procedural blunders [126].

### 5.3. Nasal Administration

Similar to the above routes, the most significant advantage of nasal administration (IN) is its ability to bypass the BBB [127]. Moreover, there is evidence that IN might be more effective than the IV route [128]. Apart from that, IN is non-invasive. Hence, repeated administration is possible, resulting in rapid distribution to the intended active site [129]. A study on mouse models with 6-OHDA-induced brain inflammation showed that sEVs are distributed throughout the brain, specifically the cerebral frontal cortex, cerebellum central, and sulcus [122]. Furthermore, a study used intranasal delivery of gold nanoparticle (GNP)-labelled MSCs-derived sEVs and X-ray computed tomography (CT) technology, which observed that sEVs administrated intranasally were retained in the brain for up to 96 h [130]. However, the intranasal route has its limitations, as only a small dosage can be administered due to small surface areas for absorption at the olfactory epithelium and a short retention time for drug absorption [127,131].

## 6. Advantages of sEVs over Stem Cells

Unlike stem cell therapy, which involves classical transplantation of live stem cells from donors into patients, sEVs are extracted from the human donor’s MSCs and sterilised. Hence, sEVs have several advantages over stem cells [132]. For neurodegenerative diseases, sEVs can cross the BBB more efficiently than MSCs [133]. By manipulating and optimising their lipids and membrane proteins, sEVs can recognise specific recipient cells to cross the BBB [134]. Studies have found that sEVs can be transfected and engineered to deliver miRNAs to the targeted neurons [135]. There are two possible mechanisms. Firstly, sEVs are internalised by endothelial cells, undergo transcytosis, and then are released to be internalised by the target recipient cells [136,137]. Alternatively, sEVs may enter the CNS through the intercellular junctions of endothelial cells [138].

Compared with stem cell therapy, using sEVs for treating neurodegenerative diseases is better tolerated in patients, improving safety. Life-threatening issues that may happen in stem cell therapy, such as immune rejection, pulmonary embolism and stress reactions resulting in the death of cells, and abnormal differentiation, can be prevented by using sEVs [132]. Furthermore, patients can be safe from unlimited cell growth and tumour development because sEVs are not constantly dividing cells. In the case of repeated injections for long-term treatment, MSC-sEVs also do not induce toxicity in patients [139]. sEVs do not undergo mutation nor induce metastasis [140]. Other than target tissues, sEVs do not form abnormal aggregations in the liver or lung, reducing toxicity risks [141]. As stem cell therapy poses risks for small vessel obstruction, sEVs have advantages including a lower immunogenicity without an obstructive vascular effect [135].

miRNAs are protected in sEVs from enzymatic degradation in biological fluids. Hence, sEVs can help miRNAs to maintain their integrity and functionality in the target neurons. This enhances the ability of sEVs to provide candidate biomarkers for investigating neurodegenerative diseases [142]. Modifications can be performed on sEVs to carry certain cargo for drug delivery. Hence, sEVs can be loaded with bioactive cargo such as miRNAs and transferred effectively to the target cells along short or long distances. sEVs have a slower clearance rate due to their protein contents, so they are more stable and can remain for longer in the brain or target sites [138]. With their unique lipid bilayer structure enveloping an aqueous core, sEVs have a remarkable ability to load themselves with both hydrophilic and hydrophobic materials, which helps introduce miRNAs to the target sites. This property also allows sEVs to effectively distribute themselves in blood and CSF and remain for a long time in systemic circulation [141]. Since sEVs can be engineered to target specific neuron populations, this opens more possibilities for developing different sEVs delivery methods, such as intravenous and intranasal administration. Hence, neurosurgical intervention can be avoided for the convenience of patients [143]. The preparation of sEVs can also be done without harmful preservatives such as dimethyl sulfoxide (DMSO), and the storage time is relatively long [139]. With a high surface/volume ratio, sEVs can efficiently transfer biomolecules to the target tissues by amplifying the ligand-gated signalling pathways [135].

## 7. Challenges and Future Perspectives

### 7.1. The Challenges and Opportunities in the Development of MicroRNA Therapeutics: A Multidisciplinary Viewpoint

In order to study and develop miRNA therapeutics, a few factors need to be further tested: pinpointing the significant miRNAs in a specific disease, delivering the miRNAs to the target area, target expression regulation, and minimising other unwanted nonspecific interactions. The complexity of miRNAs’ biology and their physicochemical nature generates many hurdles in developing miRNA therapeutics [144] (Table 3).

The majority of DNA or RNA-based modulators have a negative charge to interact with blood proteins. For example, phosphorothioate oligonucleotide is a negatively charged compound that only targets RNA to carry out hybridisation, leading to a suppression of protein expression and preventing pathological progression [145]. It possesses a considerably long half-life of 2–4 weeks. The combination of a long half-life with nonspecific binding causes sequence complementarity-dependent effects in the off-target cells [145]. Moreover, the negative charge of miRNAs also disrupts cellular entry due to the cell membranes being in negative charge. The complexity of the functions and off-target effects represent further challenges. One type of mRNA can be acted on by hundreds of miRNAs and, vice versa, one type of miRNA can target hundreds of mRNAs. This creates a problem where the identification and implementation of miR are exceedingly complicated [146]. Furthermore, miR modulators can act on the target gene expression in unrelated cell types and tissues, causing more unwanted adverse effects. Therefore, identifying miRNAs that possesses the optimal therapeutic outcome in each disease is one of the main challenges [144].

In addition, the stability of miRNA modulators and the escape of endosomes pose challenges in the development process. miRNAs and unmodified miRNA inhibitors are susceptible to degradation. Ribonuclease enzymes in the blood can break down miRNAs that occupy the cerebrovascular duplex by targeting the less stable 3’ terminus. To this end, chemical modifications have been developed. However, the modifications must be better implemented in the 5’ terminus [147]. This indicates the consequence of an asymmetry of molecules in the RNA interference activity. Other than that, intracellular trafficking takes place initially in early endosomes, which later combine with late endosomes and lysosomes with the assistance of an acidic environment [148]. However, they can be degraded by nuclease enzymes as well. Strategies related to photosensitive molecules and pH lipoplexes have been developed in order to enhance the escape of miRNAs [144].

Using sEVs as a miRNA delivery carrier also has its obstacles, including nonspecific distribution of sEVs into unrelated organs such as kidneys, lungs, pancreas, spleen, liver, etc., [149]. In recent years, some unmodified sEVs have been better collected in a specific tissue or organ than traditional drug carriers. For example, sEVs derived from M1 polarised macrophages were discovered to have a remarkable accumulation in tumour tissue, which promotes the stimulation of macrophages in the cancer tissues [150]. In another important piece of research, the accumulation of sEVs of pancreatic cancer cells (Panc-1 cell) at the target site was observed in male BALB/c nude mice; the pancreatic cancer cell sEVs showed an accumulation in tumour tissue 30 times greater than peg-pe micelles for 4 h after administration [75]. Moreover, further profound studies have demonstrated neural-derived sEVs’ isolation by immunoadsorption with L1CAM, the neuronal antibody [151]. These innovative discoveries can lead a whole new direction in the research of neurodegenerative diseases. However, the neural-derived sEVs require improvement in purification and more evaluations are needed regarding the specific surface proteins [151]. For instance, most neurodegenerative diseases jump-start or progress through neuroinflammation. The segregation of microglia-derived sEVs and the detection of abnormal inflammatory miRNAs will be integrated into the development of the advanced diagnosis of many neurodegenerative diseases [152].

In addition, sEVs tend to be cleared rapidly from the bloodstream when observed during in vivo tests, regardless of the sEVs having distinctive lipid and protein compositions comparable to phosphatidylcholine or cholesterol liposomes. No more than 5% of the injected dosage of sEVs remained in the blood for 3 h after administration [153]. This is primarily because of macrophage capture. Evidence suggests that IV injection of B16BL6 cell-derived sEVs results in rapid clearance from the blood by the macrophages originating from the liver and spleen [154]. Therefore, any development that can alter the sEVs to avoid detection by the reticuloendothelial system will notably enhance the action of sEVs to deliver miRNAs.

### 7.2. Future Perspectives

The control of miRNAs in terms of different therapy practices has progressed from bed to bench with the success of phase I and II trials. Research on miRNAs has the potential to understand the sporadic forms of AD, PD, HD, and ALS. The obstacle now is to study the position of specific miRNA and then transfer this knowledge into clinical research and development [155]. Currently, two miRNA therapeutics are being studied in vivo. miRNA-mimics and anti-miRNAs are small molecules of RNA similar to mRNA precursors and have been used to down-regulate specific target protein expression. The target can be any gene that takes part in the pathogenesis of neurodegenerative diseases or possesses a gain of function mutation. This strategy mainly relies on reducing the specific protein [148].

Moreover, vector-based, chemically modified, and “packaged” RNA oligonucleotide delivery systems have been developed in siRNAs. This knowledge can be directly translated to developing miRNA therapeutics in the future, since siRNAs and miRNAs stem from the same concept. Nevertheless, the challenge is whether these procedures will be applicable in clinical treatment due to toxicity and bioavailability. The blood-brain barrier also plays a significant part in developing drugs to act on the brain [81].

On the other hand, sEVs are part of many pathological procedures. For instance, in AD, the progression can be contributed to by sEVs through the proliferation of neurofibrillary tangles and senile plaques. sEVs are also involved in other pathogenic occurrences, such as neuroinflammation [83]. However, sEVs can bear advantageous effects regarding AD. SEVs can isolate amyloid beta and stimulate its clearance [87]. Therefore, this signifies the requirement to further study sEVs’ mechanisms and exact functions in neurodegenerative diseases. Furthermore, sEVs have been revealed in the field of biomarkers in neurodegenerative diseases [55]. In recent years, three recognisable miRNAs observed in peripheral sEVs have been presented as potential biomarkers. Even though many studies still require further validation, it is indisputable that sEVs have a high potential in diagnosing and perhaps treating neurodegenerative diseases [156].

## 8. Conclusions

miRNA-enriched sEVs from stem cells represent a future treatment to reverse neurodegenerative disease. These could help thousands of patients live a better quality of life for the rest of their lives. Despite the many advantages of using miRNA-enriched sEVs, nevertheless, many challenges need to be addressed to make this mission possible in clinical settings (Figure 6). Particular miRNAs that could be the focus of the researchers and clinicians are the following: miR-21, miR-17-92, miR-133, miR-138, miR-124, miR-146a, etc., but the relevant miRNAs are not limited to these. If any miRNAs make a breakthrough, this will be the new chapter of cell-free therapy in neurodegenerative diseases.

## Figures and Tables

**Figure 1 pharmaceutics-15-01216-f001:**
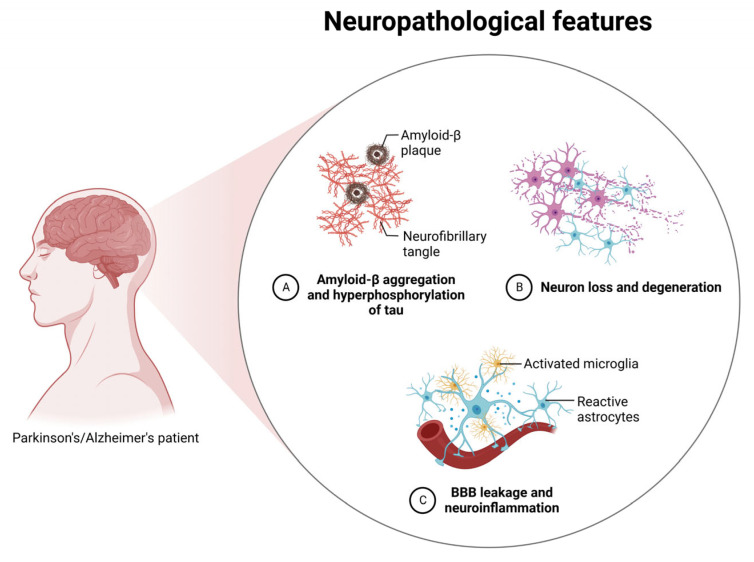
Pathophysiology of Alzheimer’s and Parkinson’s disease. Generally, the formation of amyloid-β plaques and neurofibrillary tangles results in neuron loss and degeneration. This triggers an uncontrolled immune response resulting in BBB leakage and neuroinflammation (created with BioRender.com) (accessed on 3 January 2023).

**Figure 2 pharmaceutics-15-01216-f002:**
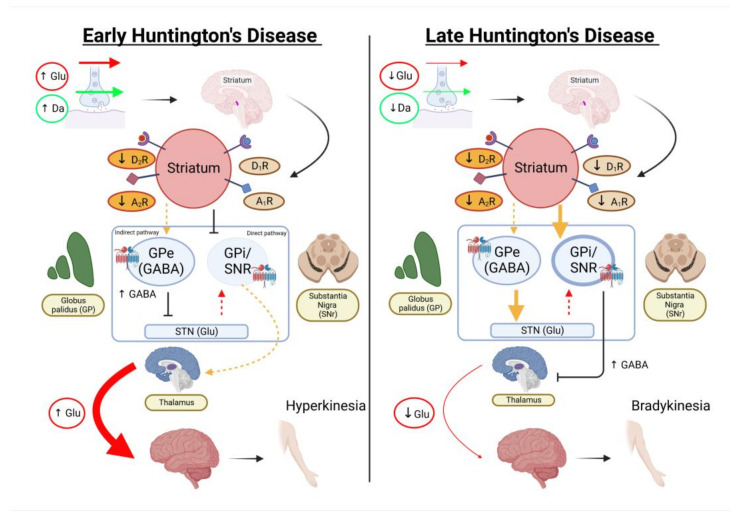
Pathophysiology of Huntington’s disease. In the early stage, the aggregates delay axonal transport, resulting in a failed delivery of GABA(A), which inhibits the synaptic excitability of the subthalamic nucleus. As a result, the compensatory mechanism of the thalamus leads to hyperkinesia. In the late stage, the trafficking of NMDA receptors in striatal neurons is affected and may cause an imbalance between synaptic and extrasynaptic NMDA receptors. This leads to excitotoxicity either due to increased glutamate or impaired uptake and clearance. Ultimately, bradykinesia is observed (created with BioRender.com) (accessed on 5 January 2023).

**Figure 3 pharmaceutics-15-01216-f003:**
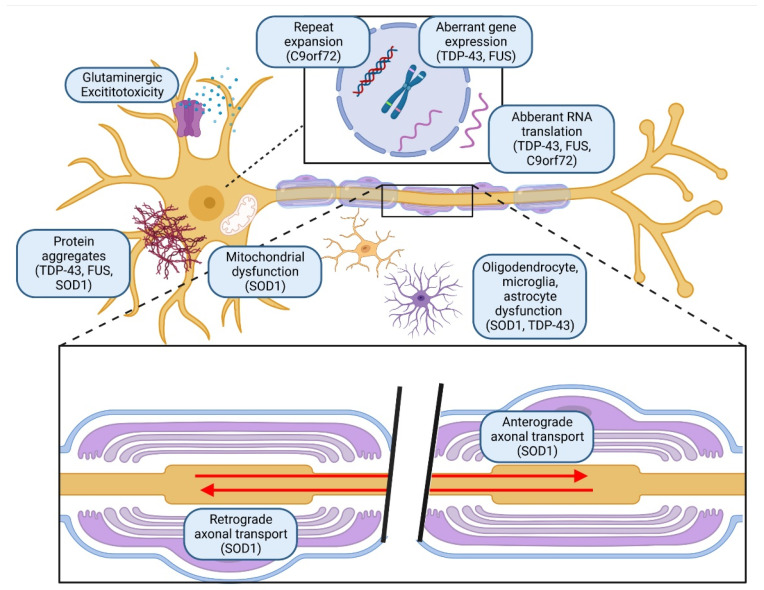
Pathophysiology of amyotrophic lateral sclerosis. The dying forward hypothesis states that ALS results in anterograde corticomotor neuron degeneration. In the dying back hypothesis, mutations in the TDP-43, c9orf72, and FUS genes form intracellular neuronal aggregates. The independent degeneration hypothesis argues that upper and lower motor neuron degeneration occurs independently (created with BioRender.com) (accessed on 8 January 2023).

**Figure 4 pharmaceutics-15-01216-f004:**
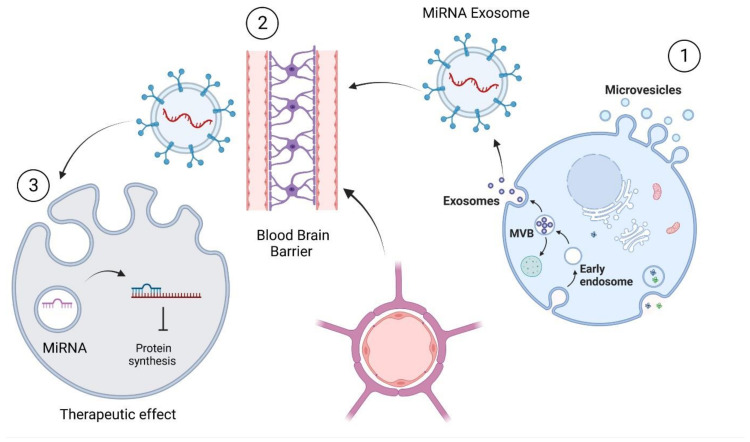
The ability of sEVs’ miRNAs to cross the blood-brain barrier presents an opportunity to correct endogenous miRNAs’ levels through miRNAs/anti-miRNAs treatment (created with BioRender.com) (accessed on 13 January 2023).

**Figure 5 pharmaceutics-15-01216-f005:**
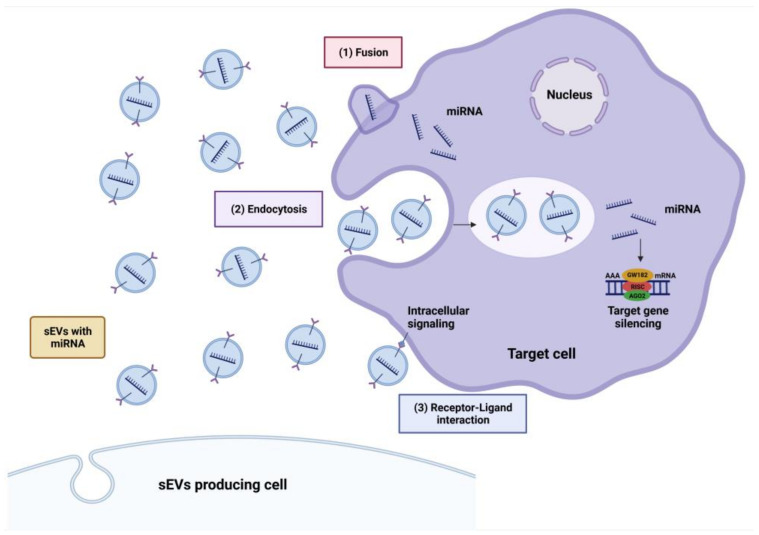
miRNA-enriched sEVs can be transmitted to target cells via three methods which are fusion, endocytosis, and receptor–ligand interactions (created with BioRender.com) (accessed on 13 January 2023).

**Figure 6 pharmaceutics-15-01216-f006:**
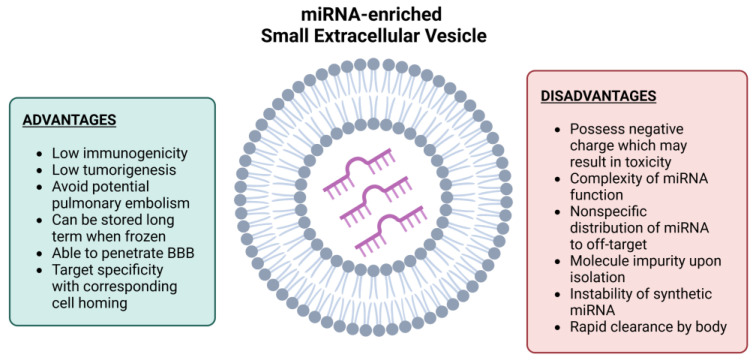
Advantages and disadvantages of miRNA-enriched small extracellular vesicles.

**Table 1 pharmaceutics-15-01216-t001:** Currently available drugs for neurodegenerative diseases.

Disease and Therapeutic Strategies	Drug Classification	Drug Example	Reference
Parkinson’s Disease Maintain CNS dopamine levels or signalling	Dopamine	Levodopa, carbidopa	[6]
	Monoamine oxidase inhibitor (type B)	Selegiline, rasagiline, safinamide	
	Catechol-O-methyltransferase inhibitors	Entacapone, Tolcapone	
	Dopamine receptor agonists	Ergot: bromocriptine Non-ergot: Ropinirole, pramipexole, rotigotine, apomorphine	[7]
	Antiviral	Amantadine	[6]
	Antimuscarinic agents	Benztropine, trihexyphenidyl	
Alzheimer’s Disease Improve cholinergic transmission within the CNS or prevent NMDA-glutamate receptor-mediated excitotoxicity	Acetylcholinesterase inhibitors	Donepezil, rivastigmine, galantamine	
	NMDA receptor antagonist	Memantine	
	Cognitive enhancers	Pyritinol, dihydroergotoxine, piribedil, citicholine, gingko biloba	[8]
	Amyloid beta-directed antibody	Aducanumab Lecanemab	[9]
Amyotrophic Lateral Sclerosis Improve survival time, slow progression of the disease	NMDA receptor antagonist	Riluzole	[6]
	Antioxidant	Edaravone	
Huntington’s Disease Symptomatic treatment to improve motor function and quality of life and suppress chorea. Does not delay the progression of the disease	Dopamine-depleting agent	Tetrabenazine	
	Dopamine antagonist	Haloperidol Olanzapine Aripiprazole	[10]
	Benzodiazepine	Clonazepam	[7]
	NMDA antagonist	Amantadine Riluzole	[6]

**Table 2 pharmaceutics-15-01216-t002:** Levels of miRNAs that are upregulated or downregulated in different diseases.

Diseases	Sample	Validated Changes		References
		Upregulated	Down-Regulated	
Alzheimer’s Disease	Plasma	N/A	miRNA 342-3p miRNA 342-3p, miRNA 125a-5p, miRNA 23a-3p	[106]
	Serum	miRNA 29a, miRNA 135a, miRNA 384, miRNA 15a-5p, miRNA 18b-5p	N/A	[107,108]
	CSF	N/A	miRNA 193b, miRNA 9-5p, miRNA 598	[109,110]
Parkinson’s Disease	Plasma	miRNA 331-5p	N/A	[111]
	Serum	miRNA 22, miRNA 23a	N/A	[107]
	CSF	miRNA 153, miRNA 409-3p	N/A	[112]
ALS	Serum	N/A	miRNA 27a-3p	[113]
	CSF	miRNA 143-5p	miRNA 132-5p, miRNA 132-3p, ex-miR-143-3p	[114]

**Table 3 pharmaceutics-15-01216-t003:** Current status in the development of miRNAs therapeutics.

miRNAs	siRNAs
Less than 20 in clinical trials	Over 60 in clinical trials
Percentage being terminated/suspended (50%)	Percentage being terminated/suspended (35%)
None in phase III trials	12% of trials in phase III

## Data Availability

Data are contained within the article.

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
