# Peer review of "Small Extracellular Vesicles’ miRNAs: Biomarkers and Therapeutics for Neurodegenerative Diseases"

_pharmaceutics, 2023, doi:10.3390/pharmaceutics15041216_

Round 1

Reviewer 1 Report

This article summarized the role and applications of miRNA-enriched exosomes for neurodegeneration diseases. Details about the pathophysiology of various neurodegeneration diseases was also provided. However, the manuscript must be significantly improved before it can be published in Pharmaceutics. Comments to the manuscript are as follows:

1.   Is the topic of the manuscript ‘miRNA’ enriched in exosomes or ‘exosomes’ enriched with miRNA? Too much background provided in the introduction section made the topic of the manuscript unclear.

2.  The source cells of exosomes have an important impact on neuron. It should be mentioned between Lines 103-119. The incomplete information may confuse the readers.

3.   Section 3.0 and 4.0, what does ‘miRNA Exosomes’ represent? ‘miRNA’ or ‘exosomes’?

4.   There are several errors in the manuscript:

Line 99-100: ‘MSC-derived Exosomes’ should be ‘MSC-derived exosomes’.

Line 346: ‘modification of the exosome cell surface protein’ should be ‘modification of the exosome with cell surface protein’.

Line 549: ‘its distribution and therapeutic and biological effects of exosomes’ should be ‘its distribution and therapeutic and biological effects’.

Line 554: ‘IV injections of exosomes could travel to the brain and improve learning and memory capabilities’ should be ‘intravenously injected exosomes could travel to the brain and improve learning and memory capabilities’.

Line 559: ‘blood circulation of exosomes is quickly cleared by macrophages’ should be ‘exosomes is quickly cleared by macrophages during circulation in blood’.

Line 603: ‘exosome molecules’ should be ‘exosomes’.

Line 607: ‘to receive their miRNA’ is hard to understand.

Line 626: ‘Exosomes are protected from being degraded by enzymes in biological fluids’ should be ‘miRNAs were protected in exosomes from enzymatic degradation in biological fluids.

5.   The reference 48 was cited wrongly.

6.   Line 397-399: The introduction of miRNA should be moved to Line 105.

7.   Abbreviations such as AD and PD should be defined the first time they are used.

Reviewer 2 Report

Specific comments

The sentence “MSC treatment involving donor cells transplantation into the patient's body often poses many challenges, as MSCs may induce immune reactions, graft versus host disease and secondary infections” is not true and should be corrected.

Due to the overlapping size range and the lack of specific markers, the current extracellular vesicle (EV) preparations including exosome preparations are highly heterogenous with undetermined purity and undefined biogenesis origin. Authors should refer to the position paper of the International Society for Extracellular Vesicles (ISEV) for guidelines on the EV nomenclature, isolation and characterization, and discuss the outstanding challenges associated with EV heterogeneity. It would therefore be more appropriate to use the term “EVs” or “small EVs” instead of “exosomes” where the purity and/or biogenesis origin are not clear.

Minimal information for studies of extracellular vesicles 2018 (MISEV2018): a position statement of the International Society for Extracellular Vesicles and update of the MISEV2014 guidelines. J Extracell Vesicles. 2018 Nov 23;7(1):1535750.

Increasingly, studies are suggesting that miRNAs in the EVs may not be present in sufficient concentration and functional configuration to elicit a timely and biologically relevant response. Authors should discuss the roles of miRNAs and proteins with reference to the below papers.

Quantitative and stoichiometric analysis of the microRNA content of exosomes. Proc Natl Acad Sci U S A. 2014 Oct 14;111(41):14888-93.

MSC exosome works through a protein-based mechanism of action. Biochem Soc Trans. 2018 Aug 20;46(4):843-853.  

MicroRNAs are minor constituents of extracellular vesicles that are rarely delivered to target cells. PLoS Genet. 2021 Dec 6;17(12):e1009951.

Reviewer 3 Report

The review from Lim et al., describing the role of miRNA-enriched MSC-derived exosomes in neurodegeneration is interesting and well organized. The authors included a brief introduction regarding neurodegenerative diseases followed by a focus on exosomal miRNA.

Anyway, i have some major concerns:

1) Starting from the title, the paper needs an extensive editing of language.
Few examples:
Title: "neurodegeneration diseases" should be changed in neurodegenerative diseases
Abstract: "Indeed, the growing ageing population will indeed...", an “indeed” should be erased

"to replace the damaged-brained cells;" brained should be replaced by brain
Chapter 3.0 "miRNA exosomes" should be replaced by Exosomal miRNA, as already used in previously published reviews

and so on. Please re-read the whole text.

2) The title is vague and generic. I would ask the authors to replace the original title to a more focused one to guide the reader from the very start of the paper.

A few examples: “Mesenchymal stem cells: a source of miRNA-enriched exosomes to treat neurodegenerative disorders” or “Mesenchymal stem cells-derived exosomal miRNAs: role in neurodegeneration”.

3) In my opinion, a small introduction on exosomes' general cargo should be added at the beginning of chapter 3.0, before focusing only on miRNAs. Previously published papers and reviews already highlighted how the whole exosomal cargo (proteins, RNA, DNA, lipids etc.) may serve both as biomarkers and treatment in various diseases. Although I do understand this is the main theme of this paper, to begin and to focus only on miRNA may be a little simplifying.

4) Although the paragraphs are, in some parts, well divided and distributed, the last part seems a little disconnected. I propose to the authors to group all the small paragraphs 6.0, 6,1, 6,2 and 6,3 in a bigger, more homogeneous one. As an alternative, to expand these small paragraphs’ content. It seems reasonable to me to keep the original title “Advantages of exosomes over stem cells”.

5) It would be very interesting in my opinion to add an additional image to summarise the main conclusion. It should comprehend both pros and cons in using miRNA-enriched exosomes.

Minor concerns:

1) It would be interesting, if possiblle, to add some specific reference in table 1, as it has been done for table 2.
2) Text in Figure 2 needs to be adjusted.

3) In figure 1, it is not clear to me the “A” text. The image shows amyloid plaques and tau NFTs, but the text is about alpha-synuclein, beta (maybe authors meant Aβ)-aggregates and plaque accumulation without any reference to tau protein.

4) In figure 5, the term “receipt cell” should be changed in “target cell”.

Round 2

Reviewer 1 Report

The authors have addressed all comments. I recommend this manuscript for publication in Pharmaceutics.

Author Response

Dear reviewer, thank you for your comments and recommend our manuscript for publication. We appreciate your time

Reviewer 2 Report

Major comments - The authors have revised the review manuscript according to the reviewer's comments. With regards to the use of the term "small EVs" instead of "exosomes" authors should cite/reference the below ISEV position papers that have given the recommended guidelines on nomenclature, isolation and characterization of EVs, particularly sEVs derived from MSCs. 

Minimal information for studies of extracellular vesicles 2018 (MISEV2018): a position statement of the International Society for Extracellular Vesicles and update of the MISEV2014 guidelines. J Extracell Vesicles. 2018 Nov 23;7(1):1535750.

Defining mesenchymal stromal cell (MSC)-derived small extracellular vesicles for therapeutic applications. J Extracell Vesicles. 2019 Apr 29;8(1):1609206.

The revised title of the revised manuscript seems inappropriate, as the review is not only focused on MSC-sEVs as therapeutics, but also covers broadly sEVs as biomarkers for neurodegenerative diseases. 

Minor comments - abbreviations should be standardized and mentioned at the beginning, and same abbreviations used thereafter throughout the review. 

Missing reference 98

Reviewer 3 Report

I thank the authors for all their answers and the changes they have made to the original text. The paper greatly improved.

I just have one last concern.
In my opinion, the new paragraph in section 3.0 should be a little wider and would benefit from the addition of 2 or 3 references. 
